# Composition-Regulated Photocatalytic Activity of ZnIn_2_S_4_@CdS Hybrids for Efficient Dye Degradation and H_2_O_2_ Evolution

**DOI:** 10.3390/molecules29163857

**Published:** 2024-08-14

**Authors:** Nikolaos Karamoschos, Andreas Katsamitros, Labrini Sygellou, Konstantinos S. Andrikopoulos, Dimitrios Tasis

**Affiliations:** 1Department of Chemistry, University of Ioannina, 45110 Ioannina, Greece; k4520fd@yahoo.gr (N.K.); andrewkatsamitros@gmail.com (A.K.); 2Foundation of Research and Technology Hellas, Institute of Chemical Engineering Sciences (ICE-HT), P.O. Box 1414, Rio, 26504 Patras, Greece; sygellou@iceht.forth.gr (L.S.); candrik@iceht.forth.gr (K.S.A.); 3Department of Physics, University of Patras, 26504 Patras, Greece; 4University Research Center of Ioannina (URCI), Institute of Materials Science and Computing, 45110 Ioannina, Greece

**Keywords:** cadmium sulfide, zinc indium sulfide, photocatalysis, characterization, dye degradation, hydrogen peroxide evolution

## Abstract

Heterostructures of visible light-absorbing semiconductors were prepared through the growth of ZnIn_2_S_4_ crystallites in the presence of CdS nanostructures. A variety of hybrid compositions was synthesized. Both reference samples and heterostructured materials were characterized in detail, regarding their morphology, crystalline character, chemical speciation, as well as vibrational properties. The abovementioned physicochemical characterization suggested the absence of doping phenomena, such as the integration of either zinc or indium ions into the CdS lattice. At specific compositions, the growth of the amorphous ZnIn_2_S_4_ component was observed through both XRD and Raman analysis. The development of heterojunctions was found to be composition-dependent, as indicated by the simultaneous recording of the Raman profiles of both semiconductors. The optical band gaps of the hybrids range at values between the corresponding band gaps of reference semiconductors. The photocatalytic activity was assessed in both organic dye degradation and hydrogen peroxide evolution. It was observed that the hybrids demonstrating efficient photocatalytic activity in dye degradation were rather poor photocatalysts for hydrogen peroxide evolution. Specifically, the hybrids enriched in the CdS component were shown to act efficiently for hydrogen peroxide evolution, whereas ZnIn_2_S_4_-enriched hybrids demonstrated high potential to photodegrade an azo-type organic dye. Furthermore, scavenging experiments suggested the involvement of singlet oxygen in the mechanistic path for dye degradation.

## 1. Introduction

Sustainability in energy issues and carbon neutrality have been largely associated with responsible management of natural resources [1]. Especially in recent decades, the quality of aqueous deposits on our planet has deteriorated due to the uncontrolled release of pollutants by industrial units. A representative example of organic pollutants is diazo-type dyes, commonly used as coloring components in textiles. The disposal of such harmful organics into aquatic environments has aroused urgent calls for the treatment of effluents containing such polluting species [2]. The target of related investigations is to succeed total mineralization of organic pollutants in aqueous systems. This may be achieved by the development of appropriate nanostructured catalytic systems that participate in various energy conversion schemes [3,4]. The ideal catalyst should have a multifunctional character and thus be involved in a variety of potential catalytic reactions [5]. Beyond the environmental remediation, the very same catalytic system may act as an efficient spark for the conversion of precursor substances to useful chemicals through redox processes [6]. One of the most widely used substances in industry as well as technology is hydrogen peroxide (H_2_O_2_), with its main applications focused on the paper industry and rocket propulsion systems [7]. Although the so-called anthraquinone method is the usual way of producing H_2_O_2_, it is an energy-demanding process that produces harmful by-products. Thus, there is an urgent need to develop multifunctional nanostructured systems for the establishment of sustainable catalyst-mediated chemical conversion protocols.

In the past few decades, the general family of advanced oxidation processes (AOPs) has been applied to the abovementioned applications [8]. These include a wide variety of mechanistic paths, based on either photocatalysis, electrochemistry, ozonation, or sonochemistry. Sunlight-driven processes have been widely considered as a renewable strategy for the successful conversion of photon energy to chemical energy. The seminal works were based on metal oxide photocatalytic systems, such as TiO_2_ and ZnO [9]. However, the optical properties of such semiconducting oxides restricted the photocatalytic activity in spectral windows of the ultraviolet region. Thus, there was a great necessity to develop alternative semiconducting species, which demonstrated appropriate absorption in the visible spectrum as well as some proper photostability. In recent years, the family of metal chalcogenides was extensively investigated for a variety of photocatalytic processes. A representative example, CdS, has shown great potential as a functional visible-light-responsive photocatalyst in a wide range of photocatalytic schemes [10]. However, the cadmium sulfide lattice demonstrated decreased structural stability due to the photocorrosion phenomena, caused by the oxidative nature of the formed holes under irradiation conditions. In recent years, the challenge of developing a functional and structurally stable photocatalytic system has emerged as a high-priority research area. When compared with the metal chalcogenide CdS, mixed metal sulfides of the general formula A_x_B_2_S_3+x_ present an enhanced photostability. A representative example of this family involves the visible-light responsive ZnIn_2_S_4_ semiconductor (ZIS) [11]. Its band gap (~2.4 eV) is appropriate for harvesting visible light photon energy, whereas the relatively large difference between the conduction band and the H^+^/H_2_ reduction potentials makes this photocatalyst an efficient system for hydrogen evolution reaction [12]. Additional applications involve the degradation of organic/inorganic pollutants [13,14] as well as CO_2_ reduction [15]. However, the performance of neat ZnIn_2_S_4_ photocatalyst is far from optimized conditions, as electron-hole recombination phenomena seem to efficiently decrease the quantum yield of photoprocesses, either reductive or oxidative ones. A simple strategy for the inhibition of such carrier recombination phenomena involves the growth of heterostructures comprising at least two components [16,17,18,19,20]. The presence of a cocatalyst species has been found to greatly catalyze the dissociation of the bound carrier pairs (excitons) through the formation of multiple heterojunctions between the semiconducting species. This may lead to subsequent reductive as well as oxidative reaction paths by electrons and holes, respectively. Efficient heterojunctions of ZnIn_2_S_4_-based systems have been recently developed for a wide range of photocatalytic processes, such as the formation of reactive oxygen species (ROS) [21], pollutant removal [22], and H_2_O_2_ evolution [23,24,25,26].

Thus, an increased focus has been given to the development of semiconducting heterojunctions with matched band gap structures. With its superior high chemical stability and low toxicity, ZIS is a ternary sulfide semiconductor that can be paired well with CdS. The band energy levels of such semiconducting species may give rise to efficient charge carrier separation [27,28,29,30]. Furthermore, the presence of ternary chalcogenide may also act as a protective layer, preventing any photocorrosion phenomena of the CdS lattice through hole-mediated oxidation [31,32].

In recent years, ZnIn_2_S_4_@CdS hybrids have been studied as potential photocatalytic systems with bifunctional performance. In certain cases, the photo-adducts were formed simultaneously. In a typical case, Zhang et al. [33] demonstrated that Z-scheme heterojunctions were able to produce 1:1 stoichiometric ratios of H_2_ and H_2_O_2_ in neat water under an inert atmosphere. It was suggested that H_2_O_2_ species evolved through a two-hole H_2_O oxidation process. The photocatalytic performance of the hybrid was enhanced by three orders when compared with the sample originating from the physical mixing of neat semiconductors. Similarly, Yang et al. [34] have developed dual-purpose ZnIn_2_S_4_@CdS hybrids for catalyzing simultaneously the H_2_ evolution and benzyl alcohol oxidation. Both H_2_ gas and benzaldehyde were found to evolve at about 1:1 stoichiometric ratio, in all studied compositions.

Independent works have performed optimization of performance in separate photocatalytic processes. Zhang and co-workers [35] studied the dual performance of ZnIn_2_S_4_@CdS hybrids for both hydrogen evolution and antibiotic removal in inert and open-air conditions, respectively. The authors concluded that the very same hybrid was the most effective in both photoprocesses, whereas the order of catalyst reactivity did not seem to depend on composition. Analogous results of hybrid catalyst photoreactivity not depending on its composition in the studied photoprocesses were demonstrated in the work of Almajidi et al. [36] In this work, both H_2_O_2_ evolution and tetracycline degradation were studied in separate photocatalytic processes.

In all the abovementioned studies [35,36], the trends of photocatalytic performances (yields) in each of the studied photoinduced reactions were not dependent on hybrid composition, a factor that did not seem to play a critical role. Specifically, if a certain hybrid at optimized composition acted as an efficient photocatalyst for one photoreaction, the very same hybrid was shown to act efficiently for the other photoinduced process. To this end, we would like to investigate a similar system, comprising ZnIn_2_S_4_ and CdS, towards its potential to function as a photocatalyst for both H_2_O_2_ evolution and an azo-dye degradation. A comparative study of neat and hybrid materials was investigated in detail, regarding their structural characterization, chemical speciation, optical properties, and photocatalytic activity. Their unique electronic properties coupled with efficient photocatalytic capabilities make these materials a multifunctional platform for environmentally friendly pollutant removal as well as the production of high-value substances.

## 2. Results and Discussion

### 2.1. Physical Properties of Semiconducting Materials

The crystallinity of either pristine semiconductors or the resulting hybrids was analyzed by the XRD technique. Regarding the neat CdS, the typical crystallographic planes of the hexagonal lattice were observed (Figure 1A). The corresponding XRD peaks range between 24.8° and 75.5° and were assigned to specific hkl Miller indices, namely (100), (002), (101), (102), (110), (103), (200), (112), (201), (004), (202), (203), (210), (211), (114), and (105), in order of increasing 2θ angle (JCPDS NO. 41-1049) [37]. The XRD pattern of the neat ZIS sample is shown in Figure 1B. The main diffraction peaks corresponded to the hexagonal crystalline phase. Specifically, certain crystallographic planes were observed at 2θ angles of 21.4° (006), 27.3° (102), 30.3° (104), 39.8° (108), 47.3° (110), 52.1° (116), 55.4° (022), and 75.8° (211), respectively (JCPDS NO. 65-2023) [36,38]. Comparison of the latter reference sample and the “ZIS 99 wt%” sample gave rise to similar patterns, due to the large excess of ZIS component (Figure 1C). In addition, the appearance of some minor peaks was observed which could be potentially ascribed to the diffraction peaks of the CdS component. One representative example is the peak at 43.6°. On the contrary, the “ZIS 16 wt%” sample demonstrated the typical diffraction peaks, assigned to CdS-based crystallographic planes as well as a broad band in the 2θ angle window between 15° and 35° (Figure 1D). The latter could be ascribed to a somewhat amorphous ZIS state within the hybrid assembly.

The morphology of either pristine semiconductors or the resulting hybrids was assessed by SEM imaging. Neat CdS sample was shown to grow in the form of dendra-like crystallites (Figure 2A) [37]. Such dendra were formed by the assembly of microstructured irregular crystals. Neat ZIS microstructures were grown in the form of quasi-spherical crystallites, with a marigold flower-like morphology. The diameter of such crystals ranged between 1.5 and 5.0 μm (Figure 2B). Concerning the “ZIS 99 wt%” sample, CdS-based nanostructures were hardly observed in most of the mapped areas, due to the low loading of metal sulfide components. The vast majority of the microstructures were composed of porous assemblies of irregularly intertwined ZIS-based leave-like structures (Figure 2C). Such porous microstructures had a wide size distribution, ranging between 1.5 and 8.5 μm. The presence of CdS crystallites was more obvious in the “ZIS 16 wt%” sample (Figure 2D). The size of the marigold-shaped crystallites ranged between 2.0 and 5.5 μm.

To investigate the chemical speciation of the samples, X-ray photoelectron spectroscopy studies were performed. Appendix A shows the survey scans from the reference mixed sulfide (neat ZIS) as well as some representative hybrids, namely “ZIS 16 wt%” and “ZIS 99 wt%” samples. As expected, the elements involved were detected in each sample, specifically Zn, In, and S. In the case of hybrid samples, the cadmium component was further recorded.

Figure 3A,B shows the XPS Zn2p and the X-ray-induced Auger (XAES) ZnL_3_M_45_M_45_ peaks from the studied samples. In Figure 3A, the Zn2p_3/2_ component was centered at 1022.3 eV [39]. No differences were observed in all studied samples, suggesting a similar chemical environment for the zinc element. In an analogous motif, the kinetic energy of the corresponding X-ray-induced Auger electron (XAES) spectra of the ZnL_3_M_45_M_45_ peak was centered at 989.5 eV in all three studied samples (Figure 3B). The modified Auger parameter, α’, defined as the sum of the binding energy of Zn2p_3/2_ (Figure 3A) and the kinetic energy of the ZnL_3_M_45_M_45_ Auger transition (Figure 3B) may provide accurate chemical state information. The aforementioned parameter was estimated to be 2011.8 ± 0.1 eV. The calculated value confirmed the presence of a Zn^2+^ valence state within the ZnIn_2_S_4_ lattice in all studied samples [40].

Appendix A shows the In3d XPS peaks from all studied samples. As observed, the In3d_5/2_ binding energy was centered at 445.1 eV. This was assigned to trivalent indium within the ZnIn_2_S_4_ lattice [41]. No apparent peak shift was observed, thus implying a similar chemical environment for the indium species in all studied samples.

Figure 3C shows the S2p doublet of a neat ZIS sample, with a spin-orbit splitting of 1.2 eV. Its S2p_3/2_ component was centered at 161.8 eV. A similar binding energy was recorded also for the “ZIS 99 wt%” sample (Appendix A). Such value confirmed the existence of mixed sulfide, ZnIn_2_S_4_ [41]. It is noted that the corresponding peak of the hybrid containing an excess of the CdS component (“ZIS 16 wt%” sample) was observed at a slightly higher binding energy, namely 162.0 eV (Appendix A) [37]. This comes in close agreement with the corresponding S2p_3/2_ peak of the reference neat CdS, which is centered at 162.1 eV (Appendix A). Thus, the slight increase in the binding energy of the “ZIS 16 wt%” hybrid may be attributed to the high CdS content (theoretical mass fraction 84 wt%), with the S2p spectrum being dominated by the CdS component.

Figure 3D shows the Cd3d XPS peaks of “ZIS 16 wt%” and “ZIS 99wt%” samples, respectively. The Cd3d_5/2_ component was centered at 405.6 eV, assigned to Cd^2+^ species within the CdS lattice [37]. This is in close agreement with the corresponding component of reference neat CdS (Appendix A). From the corresponding peak areas of In3d_5/2_, Zn2p_3/2_, S2p_3/2,_ and Cd3d_5/2_ components, the relative % atomic concentrations of the elements were derived and presented in Appendix A. It is noted that the estimated values are in close proximity to the nominal ones, in the case of neat ZIS. From the abovementioned data of XPS analysis, it was clearly shown that there is no apparent alteration of the chemical environment during the growth of the ZIS component in the presence of CdS crystals. A hypothetical alteration could be potentially associated with the generation of new species in the interface between ZnIn_2_S_4_ and CdS components.

The vibrational characterization of the studied samples was assessed by Raman analysis. The mapping of the samples took place by collection of Raman spectra from at least eight different spots of each sample. In Figure 4A, a typical Raman spectrum of neat ZIS demonstrated a group of peaks, which could be assigned to more than one species. Specifically, characteristic bands of the ZnIn_2_S_4_ lattice were observed at 251 cm^−1^, 305 cm^−1^, 345 cm^−1^, and 370 cm^−1^, respectively [42]. A second species exhibited typical bands at 154 cm^−1^, 219 cm^−1^, and 474 cm^−1^, respectively. The latter could be assigned to elemental sulfur, comprising eight-membered rings [43]. Elemental sulfur in the neat ZIS sample was not detected by XPS analysis (see above), thus we consider that the zero-valent sulfur-based species are present in trace quantities (less than 0.5 wt%).

Concerning the Raman mapping of the “ZIS 99 wt%” sample, variable information was extracted depending on the studied domain (Figure 4B,C). A general picture of the sample may be described by the recording of two different Raman spectra in two corresponding spots. The one in “yellow-green” color exhibited strong bands at 302 cm^−1^ and 603 cm^−1^, which were assigned to vibrational modes of the CdS lattice (Figure 4B) [44]. The latter spectrum was recorded by focusing on an isolated CdS grain, with a size of a few μm. In another spot of the sample, the recorded spectrum in “dark blue” showed the typical bands at 251 cm^−1^, 305 cm^−1^, 349 cm^−1^, and 370 cm^−1^, assigned previously to ZnIn_2_S_4_ (Figure 4C). This motif implies that the hybrid is in a rather phase-separated state, with low homogeneity at the micron scale. It is noted that no elemental sulfur bands were observed in all spectra collected.

On the contrary, the homogeneity was comparatively enhanced in the case of the “ZIS 16 wt%” sample. A representative Raman spectrum of the abovementioned sample is depicted in Figure 4D (violet-colored spectrum). It was shown that both semiconducting components were recorded simultaneously. Specifically, the characteristic bands of CdS (302 cm^−1^ and 603 cm^−1^) and those of ZnIn_2_S_4_ (251 cm^−1^, 350 cm^−1^, and 370 cm^−1^) were observed. No elemental sulfur bands were observed. The fact that both semiconducting components were detected in each spotting suggests enhanced uniformity down to the micron level. Furthermore, a weak broad band in the range between 400 and 450 cm^−1^ emerges in the “ZIS 16 wt%” sample. This could be ascribed to the amorphous ZnIn_2_S_4_ state [45]. This is strongly supported by the XRD data (see Figure 1D).

### 2.2. Optical Properties and Photocatalytic Activity

To establish a deeper view of the optical properties of the studied samples, diffuse reflectance spectroscopy (DRS) measurements were performed. In the inset of Figure 5A, the DRS spectrum of neat ZIS is illustrated, with the absorption edge located in the region between 450 and 550 nm. The energy band gap value of the neat semiconductor could be estimated from the tangent line in the plot of Kubelka–Munk functions against photon energy (Figure 5A). The band gap of the ZIS sample was determined to be 2.50 eV, a value that corresponds to enhanced absorbance in the visible spectrum (~496 nm). On the other hand, neat CdS exhibited an absorption edge in the region between 500 and 600 nm, with the estimated band gap being 2.23 eV (Figure 5B).

The corresponding DRS pattern of the “ZIS 99 wt%” hybrid, accompanied by the corresponding Kubelka–Munk function versus energy plot, is shown in Figure 5C. By comparing with the spectrum of neat ZIS, it was clearly observed that the band gap of the hybrid was substantially decreased to a value of 2.37 eV (523 nm). On the other hand, the “ZIS 16 wt%” hybrid demonstrated optical properties that are almost analogous to the ones of pristine CdS. The estimated energy band gap value of the abovementioned hybrid was found to be 2.27 eV (Figure 5D).

The photocatalytic activity of the materials was examined by studying both the photodegradation of a diazo-organic dye (Orange G, Appendix A) as well as the evolution of H_2_O_2_, in separate experiments. Concerning the decay kinetics of the organic dye, the monitoring of the analyte concentration was accomplished by UV-Vis absorption spectroscopy measurements (Appendix A). Specifically, the dye absorption maximum at 480 nm was followed at certain irradiation times. Besides the neat semiconductors, a great variety of hybrid compositions were tested as potential photocatalysts for dye degradation. Two different sets of samples were observed regarding the photocatalytic activity. In Figure 6A, the decay curves of the reference materials among with the most efficient hybrids are illustrated. It was clearly observed that neat CdS demonstrated poor activity for dye degradation. On the contrary, neat ZIS showed a comparatively enhanced photocatalytic activity. Further optimization could be accomplished by integrating a low % mass loading of CdS, namely 1 wt%. The latter hybrid (ZIS 99 wt%) was found to photodegrade the organic dye within about 80 min of irradiation. It is noted that three dye degradation experiments were carried out for each sample. It was observed that the maximum deviation from the mean value was 0.04. The decay curves of the less efficient compositions are shown in Appendix A. A clear trend was observed, which supports the composition-dependent photocatalytic activity. The tested samples presented photocatalytic activity for dye degradation in the following decreasing order: ZIS 99 wt% > neat ZIS ≈ ZIS 84 wt% > ZIS 96 wt% > ZIS 50 wt% > ZIS 1 wt% > ZIS 16 wt% > ZIS 4 wt% > neat CdS.

Under different conditions, the aforementioned materials were assessed as potential photocatalysts for H_2_O_2_ evolution. In oxygen-purged suspensions, the photocatalytic activity of the materials was examined by performing sampling out at specific irradiation times and subsequent titration with tetravalent ceric ions. Concerning the kinetics of H_2_O_2_ evolution, the monitoring of the Ce^4+^ concentration was accomplished by UV-Vis absorption spectroscopy measurements. Specifically, the inorganic ion absorption maximum at 316 nm was followed at certain irradiation times. Utilization of a calibration curve may match each absorption measurement with the H_2_O_2_ concentration in the irradiated suspension. As in the previous case of dye degradation, the very same samples were tested as potential photocatalysts towards the H_2_O_2_ evolution. Two different sets of samples were observed regarding the photocatalytic activity. In Figure 6B, the evolution curves of the reference materials among with the most efficient hybrids are illustrated. It was clearly observed that neat ZIS demonstrated poor activity for H_2_O_2_ evolution. On the contrary, neat CdS showed a comparatively enhanced photocatalytic activity. Further optimization could be accomplished by integrating a moderate % mass loading of ZIS, namely 16 wt%. The latter hybrid (ZIS 16 wt%) was found to photocatalyze the evolution of H_2_O_2_, reaching a yield of about 8 mM per gram of catalyst within 4 h of irradiation. It is noted that three H_2_O_2_ evolution experiments were carried out for each sample. It was observed that the maximum deviation from the mean value was ~1.5 mM/g (at t = 10 h of irradiation time). The evolution curves of the less efficient compositions are shown in Appendix A. Again, a kind of trend was observed, which supports the composition-dependent photocatalytic activity. The tested samples presented photocatalytic activity for H_2_O_2_ evolution in the following decreasing order (at t = 6 h of irradiation time): ZIS 16 wt% > ZIS 4 wt% > neat CdS > ZIS 1 wt% > ZIS 50 wt% > ZIS 84 wt% > ZIS 96 wt% ≈ ZIS 99 wt% ≈ neat ZIS.

The cycling stability of the optimized hybrid for dye degradation was assessed by consecutive dye degradation tests (Figure 6C). It was demonstrated that the performance of the hybrid in the second cycle was somewhat decreased due to potential physical adsorption phenomena onto the catalyst surface. It is noted, though, that the performance remained stable in the third and fourth cycles, respectively. Concerning the dye degradation study by ZIS-enriched hybrids, the photocatalytic process took place in the presence of various scavengers of transient species (Figure 6D). By following the corresponding decay profiles, one may draw useful conclusions about the mechanistic path through which the dye degradation takes place in the presence of the “ZIS 99 wt%” sample. A rather weak inhibition of the latter process was observed in the case where isopropanol was used as a quencher. It is noted that the alcohol is used as a hydroxy radical scavenger [37]. A more pronounced inhibition of the decay process was demonstrated in the case where sodium azide was used as a quencher. The inorganic species is responsible for the scavenging of both hydroxy radicals and singlet oxygen [37]. From the observed profiles, we strongly suggest that the degradation of diazo-derivative takes place mainly through the generation of excited oxygen species. We suggest that in the case of the “ZIS 99 wt%” sample, the ZnIn_2_S_4_ component is exclusively excited due to its large excess. The formation of singlet oxygen species takes place through an energy transfer mechanism between the ZIS excited state and the ground state molecular oxygen. Thus, the oxidative degradation of organic dye occurs via the singlet oxygen species, a suggestion that is strongly supported by scavenging experiments. Regarding the H_2_O_2_ evolution by the optimized photocatalytic system (ZIS 16 wt%), it is suggested that both semiconducting components are excited simultaneously. In the presence of isopropanol as a hydroxy radical scavenger, an appreciable inhibition of H_2_O_2_ evolution was observed (Appendix A). Thus, we suggest that photoinduced synthesis of peroxide takes place mainly through hydroxy radicals.

Our results were compared with analogous photocatalytic systems, based on the ZIS component. Regarding the degradation of organic substances, the **t_50_** index was given for comparison (Appendix A). This involves the time needed for the analyte concentration to reach half of its initial value. Furthermore, the H_2_O_2_ yield (in mM/g/h) was given for the selected works. The abovementioned data demonstrated a wide scattering of values, since the photocatalytic activity depends on a variety of factors, such as emission spectrum and intensity of light sources, catalyst concentration, the presence of scavenging species, etc.

## 3. Materials and Methods

### 3.1. Reagents

The following reagents were purchased by Sigma Aldrich and were used as received: Cadmium Chloride (CdCl_2_), thiourea SC(NH_2_)_2_, Zinc Nitrate hexahydrate (Zn(NO_3_)_2_.6H_2_O), Thioacetamide C_2_H_5_NS, Indium Chloride InCl_3_, Cerium Sulfate Ce(SO_4_)_2,_ Sulfuric acid H_2_SO_4_, Orange G, ethanol, 2-propanol, and sodium azide (NaN_3_).

### 3.2. CdS Synthesis

For the synthesis of CdS, a slightly modified synthetic protocol of Paraschoudi et al. work was adopted [37]. Specifically, 202 mg thiourea and 242 mg CdCl_2_ with 12 mL deionized water were added to a glass vial. After ultrasonic treatment for 1 min, the aqueous solution was transferred into a Teflon-lined stainless-steel autoclave and heated in an oven at a temperature of 210 °C for 4 h. Then, the autoclave was left to cool down at room temperature and the resulting suspension was centrifuged at 6000 rpm and washed with deionized water and methanol three times each. Lastly, the isolated solid was dried at 90 °C overnight.

### 3.3. ZIS Synthesis

For the synthesis of ZIS, 60 mg Zn(NO_3_)·6H_2_O, 88.4 mg InCl_3_, and 60 mg TAA with 12 mL deionized water were added to a glass vial [13]. After ultrasonic treatment for 1 min, the aqueous solution was added into a Teflon-lined stainless-steel autoclave and heated in an oven at a temperature of 120 °C for 10 h. Then, the autoclave was left to cool down at room temperature and the resulting suspension was centrifuged at 6000 rpm and washed with deionized water and methanol three times each. Lastly, the isolated solid was dried at 90 °C overnight.

### 3.4. Synthesis of CdS@ZnIn_2_S_4_ Hybrids

The protocol for the synthesis of CdS@ZnIn_2_S_4_ hybrids is similar to the one described in Part 3.3. The only difference is that appropriate amounts of CdS material are mixed with the precursor substances for ZIS growth. The theoretical weight fractions of ZIS component are 1 wt%, 4 wt%, 16 wt%, 50 wt%, 84 wt%, 96 wt%, and 99 wt%, respectively. For simplicity, each sample was denoted as “ZIS x wt%”, with the x value indicating the mass loading of the ZIS component in the hybrid.

### 3.5. Characterization

Powder XRD measurements were recorded using a BRUKER AXS (D8 ADVANCE, Bruker, Billerica, MA, USA) unit equipped with a Cu X-ray tube. The morphology of the hybrids was assessed by Scanning Electron Microscopy (SEM) imaging (model JSM-6510LV) (UoI microscopy unit).

For the study of the structure of the materials at the molecular level, the micro-Raman setup T-64000 JY (Horiba group) was used. Samples were illuminated by 514.5 nm light emitted from an SSD laser which was focused by a 50× long working distance microscope objective. The backscattered light was directed to the entrance slit of a single-stage monochromator after passing through an appropriate edge filter which rejected the Rayleigh scattered photons. The monochromator spectral resolution using the 600 gr/mm grating was ~6 cm^−1^ on average throughout the whole spectral range covered in the experiments. Detection was accomplished by a 2D CCD cooled down to 132 K using liquid nitrogen.

The photoelectron spectroscopy experiments were carried out in an ultra-high vacuum system (UHV) which was equipped with an X-ray gun and a SPECS Phoibos 100-1D-DLD energy analyzer. An unmonochromatized MgKα line at 1253.6 eV and an analyzer pass energy of 15 eV (giving a full width at half maximum (FWHM) of 0.85 eV for the Ag3d_5/2_ peak) were used. The XPS core level spectra were analyzed using a fitting routine, which can decompose each spectrum into individual mixed Gaussian–Lorentzian peaks after a Shirley background subtraction. Errors in our quantitative data are found in the range of ~10%, (peak areas) while the accuracy for the BE assignments is ~0.1 eV. The samples were in powder form and pressed into pellets, and the analyzed area was a rectangle of 7.0 × 10 mm^2^.

The UV–Vis diffuse reflectance spectra (DRS) of the fabricated catalyst (powder) were recorded using a Shimadzu 2600 spectrophotometer bearing an IRS-2600 integrating sphere (Shimadzu, Kyoto, Japan) in the wavelength of 200–800 nm at room temperature, using BaSO_4_ (Nacalai Tesque, extra pure reagent, Kyoto, Japan) as a reference sample.

### 3.6. Photodegradation of Orange G Dye

For the prosecution of the photocatalytic tests, 50 mg of either pure photocatalyst or hybrid was added to 99 mL of ultrapure water in a 250 mL round-bottomed flask. The solution was sonicated in the dark for 30 min. At the same time, the solution of the dye Orange G is prepared. In a 50 mL glass bottle, 10 mg of Orange G is added with 10 mL of ultrapure water in order to create a 1 mg/1 mL Orange G solution. The dye solution is also placed for stirring in the dark for 30 min. Then, the photocatalyst mixture was transferred into a 250 mL reactor (Lenz, Germany) and 1 mL of the Orange G solution was added. The reactor was placed in the solar simulator with 2 lamps of 800 W each as the excitation source. Continuously, the reactor with the suspension was kept under stirring in the dark for 30 min. In addition, to keep the temperature around 23 °C we used a tap water cooling circuit. Then, at specific times (0, 15, 30, 45, 60, 90, and 120), aliquots of 3 mL were taken out of the reactor with a 10 mL syringe and stored in a glass vial in a dark place. The samples were each time filtered (through 0.22 μm PTFE syringe filters) and their absorbance at 480 nm was measured using a visible absorption spectrophotometer and the relative concertation (C/C_0_) was plotted against time.

### 3.7. Photocatalytic Evolution of Hydrogen Peroxide

For a typical experiment, in a round bottom flask, we added 50 mg of the photocatalyst and 99 mL of deionized water and sonicated for 30 min. Then, the suspension was transferred into a 250 mL reactor (Lenz, Germany) with 1 mL ethanol. Continuous bubbling with oxygen for 10 min was followed by continuous stirring, and a tap water circuit was used to keep the temperature around 27 °C. Furthermore, we started the suntest with 2 lamps of 800 w each, and samples were collected at predetermined time intervals (0, 1, 2, and 4 h). Moreover, 1 mL samples were taken with a 10 mL syringe and filtered through 0.22 μm PTFE syringe filters. Then, 3 mL of 1 mM Ce^4+^ stock solution was added to each sample, and the absorption of Ce^4+^ at 316 nm was measured with a UV-Vis spectrophotometer. Finally, the H_2_O_2_ concentration was determined by the standard calibration curve that we constructed.

## 4. Conclusions

This study provides a simple approach to developing heterojunctions of visible light-absorbing semiconductors through a two-step hydrothermal process. This is the first work to demonstrate that photocatalytic performance may be tuned by component composition. Specifically, the hybrids enriched in the CdS component were shown to act efficiently for hydrogen peroxide evolution, whereas ZnIn_2_S_4_-enriched hybrids demonstrated high potential to photodegrade an azo-type organic dye. The cycling stability of the materials is considered adequate after four consecutive steps of the dye degradation process.

## Figures and Tables

**Figure 1 molecules-29-03857-f001:**
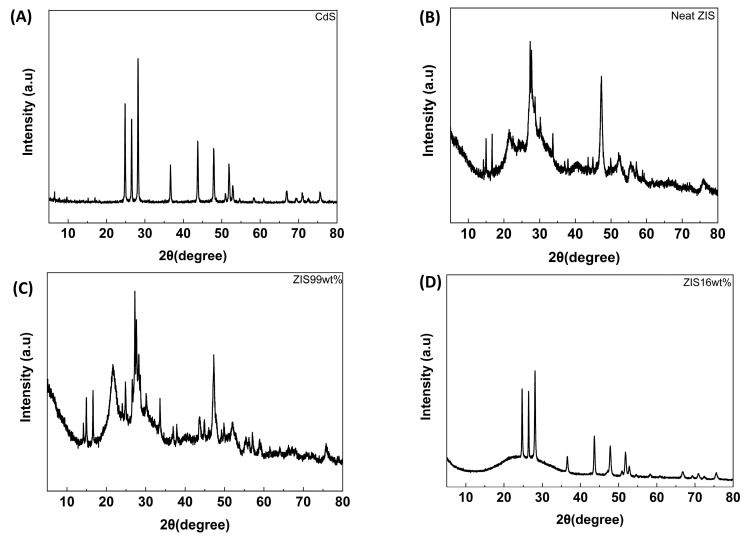
X-ray diffractograms of (**A**) neat CdS, (**Β**) neat ZIS, (**C**) “ZIS 99 wt%” hybrid, and (**D**) “ZIS 16 wt%” hybrid.

**Figure 2 molecules-29-03857-f002:**
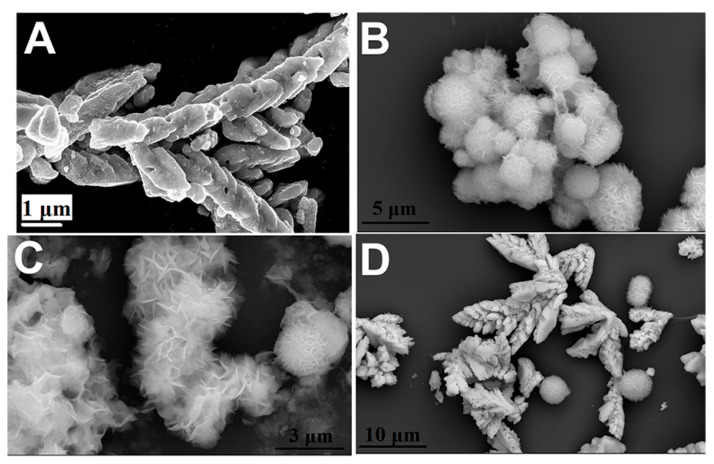
SEM imaging of (**A**) neat CdS, (**B**) neat ZIS, (**C**) ZIS 99 wt%, and (**D**) ZIS 16 wt%.

**Figure 3 molecules-29-03857-f003:**
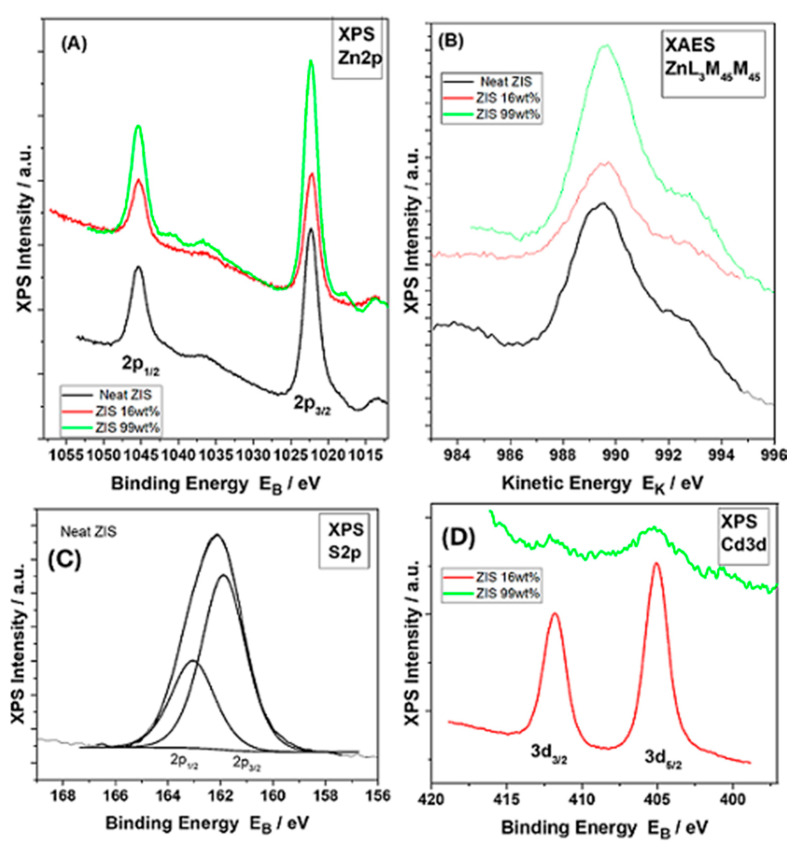
(**A**) Zn2p XPS core level peaks of neat ZIS, ZIS 99 wt%, and ZIS 16 wt% samples; (**B**) ZnL_3_M_45_M_45_ XAES peaks of neat ZIS, ZIS 99 wt% and ZIS 16 wt% samples; (**C**) S2p XPS core level peaks of neat ZIS; (**D**) Cd3d XPS core level peaks of ZIS 99 wt% and ZIS 16 wt% samples.

**Figure 4 molecules-29-03857-f004:**
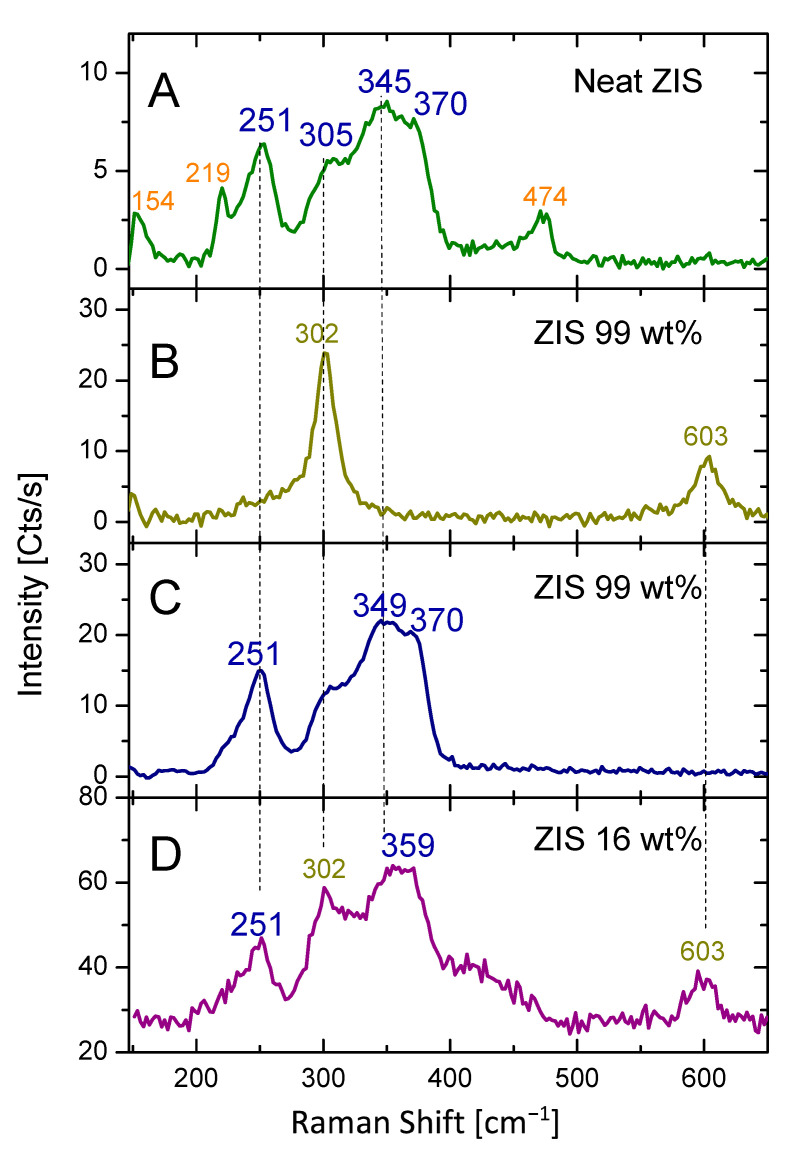
Raman analysis of (**A**) neat ZIS, (**B**,**C**) “ZIS 99 wt%”, and (**D**) “ZIS 16 wt%” samples.

**Figure 5 molecules-29-03857-f005:**
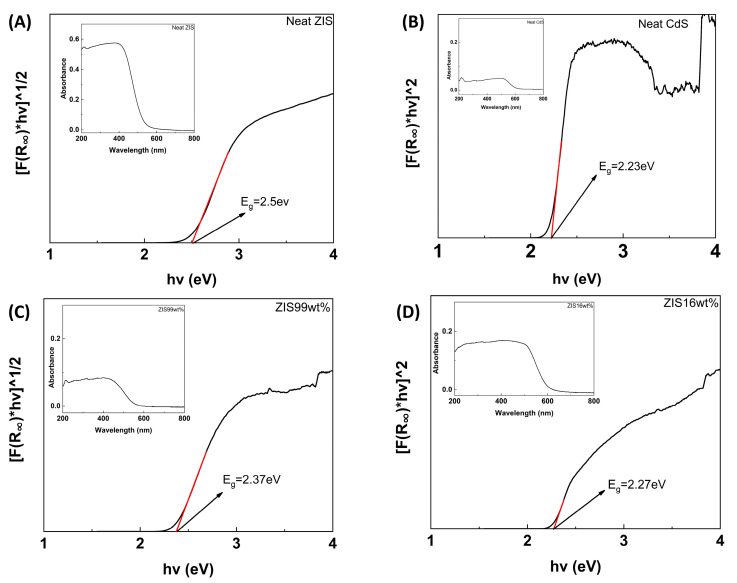
Estimation of band gap via the Kubelka–Munk function and DRS data (insets) of (**A**) neat ZIS; (**B**) neat CdS; (**C**) “ZIS 99 wt%” hybrid; and (**D**) “ZIS 16 wt%” hybrid.

**Figure 6 molecules-29-03857-f006:**
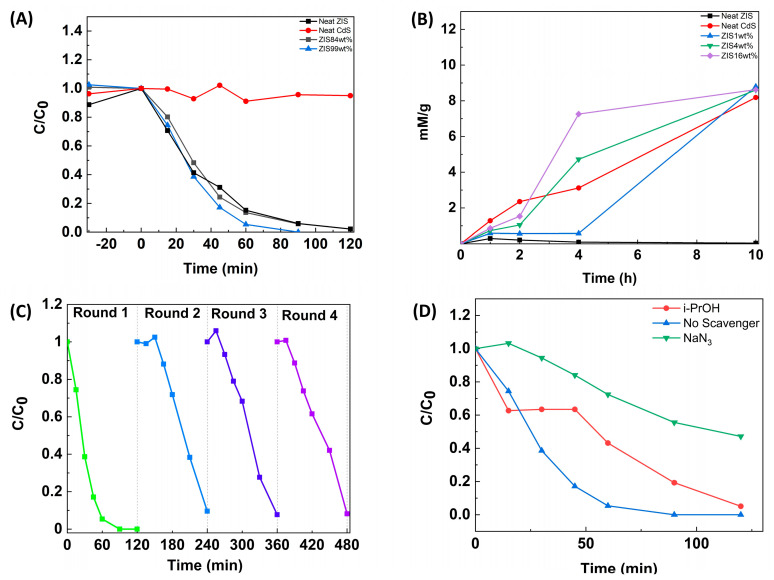
(**A**) Dye degradation profiles for neat CdS, neat ZIS, ZIS 84 wt%, and ZIS 99 wt% samples; (**B**) hydrogen peroxide evolution profiles for neat CdS, neat ZIS, ZIS 1 wt%, ZIS 4 wt% and ZIS 16 wt% samples; (**C**) consecutive cycles of dye photodegradation experiments of the ZIS 99 wt% sample; (**D**) scavenging experiments for dye degradation experiments in the presence of ZIS 99 wt% sample.

## Data Availability

The data presented in this study are available on request from the corresponding author.

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
