# Peer review of "Composition-Regulated Photocatalytic Activity of ZnIn2S4@CdS Hybrids for Efficient Dye Degradation and H2O2 Evolution"

_molecules, 2024, doi:10.3390/molecules29163857_

Round 1

Reviewer 1 Report

Comments and Suggestions for Authors

In this work, ZnIn2S4@CdS hybrids were constructed through the growth of ZnIn2S4 crystallites in the presence of CdS nanostructures for efficient photocatalytic dye degradation and H2O2 evolution. ZIS 99wt% presented the best photocatalytic activity for dye degradation and ZIS 16wt% for H2O2. The authors conducted certain characterizations of catalysts and proposed the possible reaction mechanism for dye degradation. However, the microstructure of ZnIn2S4@CdS remains ambiguous, including the specific morphology of ZnIn2S4 grew on CdS. The explanation for the photocatalytic mechanism of ZnIn2S4@CdS hybrids is not mentioned. The logical structure of the manuscript is somewhat disorganized and its data processing is rudimentary. The manuscript also contains many formatting and grammatical errors. Hence, I am unable to recommend its publication and some comments are for your reference.

1.     Transmission electron microscopy test is recommended to reveal the specific morphology of ZnIn2S4 grew on CdS.

2.     In XRD patterns, the XRD standard cards of neat CdS and ZIS should be added.

3.     The manuscript lacks an XPS analysis of Cd3d and S2p for neat CdS, and in Figure S4, the author explained the slightly higher binding energy at 162.0 eV was caused of different composition of the hybrid samples, it may need to be compared to S2p of neat CdS.

4.     In Figure 4B, the “yellow green” color assigned to vibrational modes of CdS lattice should better be drew in another new figure.

5.     It is not reasonable to estimate the band gap values of “ZIS 99wt%” and “ZIS 16wt%”, and the Y-axis should be scaled and valued in Figure 5a and b.

6.     In Figure 6a, before irradiation, the adsorption of dye should reach equilibrium, so the specific time “-30 and -5 min” should be added.

7.     In Figure 6b, why could H2O2 be detected in all samples at 0 h?

8.     The mechanism of photocatalytic H2O2 production is not mentioned, and error bars are required in all photocatalytic activity tests.

9.     There are many problems of irregular format in the manuscript, such as the letters in the figure number are lowercase in Figure 4 and Figure 5, which are not matched the notes.

Comments on the Quality of English Language

Needs to be improved

Author Response

Author responses to Reviewer1 queries

We would like to acknowledge the fruitful comments of Reviewer 1, which enhanced the quality of the manuscript. Below, the queries are illustrated with the corresponding author response. Any addition/correction in the manuscript is in red color.

In this work, ZnIn2S4@CdS hybrids were constructed through the growth of ZnIn2S4 crystallites in the presence of CdS nanostructures for efficient photocatalytic dye degradation and H2O2 evolution. ZIS 99wt% presented the best photocatalytic activity for dye degradation and ZIS 16wt% for H2O2. The authors conducted certain characterizations of catalysts and proposed the possible reaction mechanism for dye degradation. However, the microstructure of ZnIn2S4@CdS remains ambiguous, including the specific morphology of ZnIn2S4 grew on CdS. The explanation for the photocatalytic mechanism of ZnIn2S4@CdS hybrids is not mentioned. The logical structure of the manuscript is somewhat disorganized and its data processing is rudimentary. The manuscript also contains many formatting and grammatical errors. Hence, I am unable to recommend its publication and some comments are for your reference.

Query1.     Transmission electron microscopy test is recommended to reveal the specific morphology of ZnIn2Sgrew on CdS.

Response: Unfortunately, there is no availability of TEM facility in our affiliation.

Query2.     In XRD patterns, the XRD standard cards of neat CdS and ZIS should be added.

Response: The JCPDS data have been added in the manuscript (see first paragraph of Results and Discussion part).

Query3.     The manuscript lacks an XPS analysis of Cd3d and S2p for neat CdS, and in Figure S4, the author explained the slightly higher binding energy at 162.0 eV was caused of different composition of the hybrid samples, it may need to be compared to S2p of neat CdS.

Response: The requested XPS data for neat CdS have been inserted in Supplementary Information, as Figure S5. The discussion in the manuscript has been revised accordingly in the end of Page 6.

Query4.     In Figure 4B, the “yellow green” color assigned to vibrational modes of CdS lattice should better be drew in another new figure.

Response: Figure 4 has been revised accordingly. In Figure 4B, the Raman spectrum of CdS component of the “ZIS 99 wt%” sample is illustrated. In Figure 4C, the Raman spectrum of ZIS component of the “ZIS 99 wt%” sample is illustrated. In Figure 4D, the Raman spectrum of the “ZIS 16 wt%” sample is illustrated. The caption has been analogously modified. Some minor changes have been done in the paragraph below the Figure.

Query5.     It is not reasonable to estimate the band gap values of “ZIS 99wt%” and “ZIS 16wt%”, and the Y-axis should be scaled and valued in Figure 5a and b.

Response: We consider that it makes no difference adding scales and values in Y-axis. The majority of related articles in the literature present analogous spectra without adding scaling in Y-axis. The band gap value is calculated from the intersection of tangent with X-axis.

Query6.     In Figure 6a, before irradiation, the adsorption of dye should reach equilibrium, so the specific time “-30 and -5 min” should be added.

Response: In both Figure 6A and Figure S7, the absorbance data in the 30 min “dark” period were added.

Query7. In Figure 6b, why could H2O2 be detected in all samples at 0 h?

Response: We agree with the Reviewer. We corrected the kinetic curves in Figure 6B, so that no H2O2 was detected at t=0. Same was applied in Figure S8 (see Supplementary Information).

Query8. The mechanism of photocatalytic H2O2 production is not mentioned, and error bars are required in all photocatalytic activity tests.

Response: Some revision has been carried out in the part of Results-Discussion about the potential mechanistic paths (shown below), whereas Figure S9 has been added in Supplementary Information:

From the observed profiles, we strongly suggest that the degradation of diazo-derivative takes place mainly through the generation of excited oxygen species. We suggest that in the case of “ZIS 99 wt%” sample, the ZnIn2S4 component is exclusively excited due to its large excess. The formation of singlet oxygen species takes place through energy transfer mechanism between the ZIS excited state and the ground state molecular oxygen. Thus, the oxidative degradation of organic dye occurs via the singlet oxygen species, a suggestion which is strongly supported by scavenging experiments. Regarding the H2O2 evolution by the optimized photocatalytic system (ZIS 16 wt%), it is suggested that both semiconducting components are excited simultaneously. In the presence of isopropanol as hydroxy radical scavenger, an appreciable inhibition of H2O2 evolution was observed (Figure S9). Thus, we suggest that photoinduced synthesis of peroxide takes place mainly through hydroxy radicals.

Regarding the addition of error bars in all photocatalytic tests, we faced an issue concerning the somewhat overlap of certain curves in either Figure 6A or Figure 6B. For each sample, three photocatalytic experiments were carried out. In the dye degradation experiments, the maximum deviation from the mean value was 0.04. In the H2O2 evolution experiments, the maximum deviation from the mean value was ⁓1.5 mM/g (at t = 10 h irradiation time). The abovementioned data have been added in the related discussion, within the manuscript.  

Query9.     There are many problems of irregular format in the manuscript, such as the letters in the figure number are lowercase in Figure 4 and Figure 5, which are not matched the notes.

Response: All the lowercase letters in either captions or Figures (specifically Figs 1, 5 and 6) have been modified to uppercase letters, in close agreement with the text.

Reviewer 2 Report

Comments and Suggestions for Authors

Dear editor, dear authors,

After revision by the authors, the article can be published.

The article " Composition-Regulated Photocatalytic Activity of 2 ZnIn2S4@CdS Hybrids for Efficient Dye Degradation and H2O2 3 Evolution " is dedicated to the current topic of water purification from organic compounds by the catalytic process of decomposition of the dye into simpler components. The work is good, the experimental data are presented simply and clearly.

But, there are several missed points, answers to which will decorate and make the article more comprehensively perceived.

1)    The formula and optical absorption spectrum of the dye solution before and after degradation should be given.

In this regard, the question is: what does the dye decompose into during the catalytic reaction?

2)    There are no comparisons with the work of other authors in this area.

Let's note a few typos in the design.

1) Double numbering of the bibliography

2) In the caption of Fig. 1, lowercase letters are used -a, b, c... In the caption of Fig. 2 - capital A, B, ..

3) Сorrect the caption the Table S1.

3) Usually the Materials and Methods part is located after the introduction, etc.

These wishes do not beg the merits of the authors, and the article represents good experimental material and can be published after minor revision.

Author Response

Author responses to Reviewer2 queries

We would like to acknowledge the fruitful comments of Reviewer 2, which enhanced the quality of the manuscript. Below, the queries are illustrated with the corresponding author response. Any addition/correction in the manuscript is in red color.

Dear editor, dear authors,

After revision by the authors, the article can be published.

The article "Composition-Regulated Photocatalytic Activity of ZnIn2S4@CdS Hybrids for Efficient Dye Degradation and H2O2 Evolution" is dedicated to the current topic of water purification from organic compounds by the catalytic process of decomposition of the dye into simpler components. The work is good, the experimental data are presented simply and clearly.

But, there are several missed points, answers to which will decorate and make the article more comprehensively perceived.

Query1.    The formula and optical absorption spectrum of the dye solution before and after degradation should be given. In this regard, the question is: what does the dye decompose into during the catalytic reaction?

Response: The formula and the corresponding absorption spectra were inserted in a newly added Figure of Suppl. Info. (Figure S6). Regarding the decomposition adducts and the potential mechanistic paths of dye degradation, it was out of the scope of the present study. Mass spectra should be recorded, a facility which was not available in our institution.  

Query2.    There are no comparisons with the work of other authors in this area.

Response: In the end of Results and Discussion part, we added the following text:

Our results were compared with analogous photocatalytic systems, based on ZIS component. Regarding the degradation of organic substances, the t50 index was given for comparison (Table S2). This involves the time needed for the analyte concentration to reach the half of its initial value. Furthermore, the H2O2 yield (in mM/g/h) was given for the selected works. The abovementioned data demonstrated a wide scattering of values, since the photocatalytic activity depends on a variety of factors, such as emission spectrum and intensity of light sources, catalyst concentration, presence of scavenging species etc.

We added some related literature data in Table S2 (Suppl. Info.), which is a newly added Table.

Below, we have corrected some additional points, raised by Reviewer:

Point1. Double numbering of the bibliography

Response: It is corrected.

Point2. In the caption of Fig. 1, lowercase letters are used -a, b, c... In the caption of Fig. 2 - capital A, B, ..

Response: All the lowercase letters in either captions or Figures (specifically Figs 1, 5 and 6) have been modified to uppercase letters, in close agreement with the text.

Point3. Сorrect the caption the Table S1.

Response: The caption has been revised.

Point4. Usually the Materials and Methods part is located after the introduction, etc.

These wishes do not beg the merits of the authors, and the article represents good experimental material and can be published after minor revision.

Response: Recent publications of Molecules journal (of the current year 2024) have a specific template, with the following order: Introduction – Results and Discussion – Materials and Methods – Conclusions, etc.

We, thus, used this template.

Round 2

Reviewer 1 Report

Comments and Suggestions for Authors

The manuscript has been improved and publication is suggested.